# Efficacy of Two Probiotic Products Fed Daily to Reduce *Clostridium perfringens*-Based Adverse Health and Performance Effects in Dairy Calves

**DOI:** 10.3390/antibiotics11111513

**Published:** 2022-10-29

**Authors:** Charley Cull, Vijay K. Singu, Brooke J. Cull, Kelly F. Lechtenberg, Raghavendra G. Amachawadi, Jennifer S. Schutz, Keith A. Bryan

**Affiliations:** 1Midwest Veterinary Services, Inc., Oakland, NE 68045, USA; 2Central States Research Centre, Inc., Oakland, NE 68045, USA; 3Department of Clinical Sciences, College of Veterinary Medicine, Kansas State University, Manhattan, KS 66506, USA; 4Chr. Hansen, Inc., Milwaukee, WI 53214, USA

**Keywords:** calves, *Clostridium perfringens*, direct-fed microbials, health, performance, probiotics

## Abstract

*Clostridium perfringens* is a spore-forming, anaerobic bacterium which produces toxins and exoenzymes that cause disease in calves, especially necro-hemorrhagic enteritis-associated diarrhea often resulting in death. *Clostridium* infections are currently being treated with antibiotics, but even with the prudent administration of antibiotics, there are significant rates of recurrence. Probiotics, an alternative to antibiotics, are commonly employed to prevent clostridial infections. The objectives of our study were to demonstrate that two commercially available products, when used as daily, direct-fed microbials, are effective in reducing adverse effects of an experimentally induced *C. perfringens* infection in dairy calves. We conducted a single site efficacy study with masking using a randomized design comprising 10 calves allocated to 3 treatment groups (probiotic 1, probiotic 2, and control). The procedures such as general health scores, body weight, blood samples, and fecal sample collections were done followed by experimental challenge of calves with *C. perfringens*. Daily feeding of *L. animalis* LA51 and *P. freudenreichii* PF24 without or with *Bacillus lichenformis* CH200 and *Bacillus subtilis* CH201, before, during and after an oral challenge of *C. perfringens* significantly reduced the incidence and severity of diarrhea while improving general impression and appearance scores of calves. Most notably, survival of calves in the two probiotic-fed groups was significantly higher than for control calves and further substantiates the potential economic and health benefits of feeding effective probiotics.

## 1. Introduction

*Clostridium perfringens* is a rod-shaped, Gram-positive, spore-forming, anaerobic bacterium which produces toxins and exoenzymes that cause diseases [1,2]. The toxinotype (A, B, C, D, and E) classification of *C. perfringens* strains are based off which major toxins (α, β, ε, and ι) the strains produce [3]. The most common toxin produced by *C. perfringens* is the α-toxin because it is produced by each strain of the bacterium. Though it is not responsible for the virulence of *C. perfringens* [4], it is thought to be important for its pathogenesis, nonetheless. One study found a direct correlation between the amount of α-toxin in the intestine of a broiler chicken and the severity of intestinal lesions found in broiler chickens inoculated with a mutant α-toxin strain [5]. Furthermore, α-toxin is important for gas gangrene in humans and necro-hemorrhagic enteritis in bovine which supports that the toxin could cause hemorrhage, myonecrosis, and neutrophil infiltration in mammals [6,7]. Gastrointestinal diseases can also be caused by *C. perfringens* due to their enterotoxins. One of the most common is enterotoxemia which occurs when the enterotoxins circulate throughout the body, damaging tissues and organs including the brain [8]. Furthermore, allergies and gastrointestinal infections are potentially connected to this opportunistic pathogen [8,9]. Microbiota dysbiosis-associated diarrhea, which occurs frequently with the use of antibiotics, is most often caused by *C. difficile* [10]. *C. difficile* is also responsible for the most commonly reported nosocomial infections: *C. difficile* infections [11]. Therefore, new methods of prevention and treatment are being considered for humans which include bacteriocins, bacteriophages, fecal microbiota transplantation, new antibiotics, and probiotics [12].

In addition, alternatives for disease prevention in animals are being considered due to the European ban of antibiotics in livestock. One alternative of particular interest is daily feeding of probiotics [13,14]. Probiotics are being considered because a host’s susceptibility to disease is heavily influenced by its microflora which can benefit or harm the health of its intestines [15]. *Escherichia coli*, *Lactobacillus* spp., and *Streptococcus* spp. are a part of the normal microflora of the small intestine [16], and some of these bacteria, such as *Lactobacillus* spp. and *Bifidobacterium* spp., prove to be beneficial to the host when maintained in the small intestine making them probiotics of specific interest [17]. According to a previous study, probiotics have showed varied performances against Clostridia infections [18]. Lactic acid bacteria (LAB) populations are reportedly able to be maintained and even increased in the intestines by feeding probiotics and synbiotics [19]. More specifically, *Bifidobacterium* spp., *Lactobacillus* spp., and *B. subtilis* are capable of maintaining beneficial bacterial populations within the intestine [20]. *B. subtilis* is used in the feed industry [21] because it promotes beneficial microflora changes in the intestines [22], diarrheal recovery [23], and improved average daily gain and feed efficiency [24]. *Lactobacillus* spp., which are already members of the intestinal microflora and which are capable of preventing *C. perfringens* from colonizing in the intestines, have been used to treat necrotic enteritis (NE) in poultry [25,26]. *L. fermentum* strain 104R was able to eradicate *C. perfringens* β2 production in an in vitro system by decreasing environmental pH which consequently also decreased cpd2 mRNA [27]. Many researchers have been focusing on lactic acid bacteria in recent studies due to their ability to generate antagonistic metabolites such as bacteriocins [23] (Jack et al., 1995). Lactic acid bacteria can also produce other metabolites such as carbon dioxide, diacetyl, hydrogen peroxide [28,29], and organic acids [30]. Despite these studies, *Lactobacillus* spp. effect on inflammation caused by α-toxin and *C. perfringens* is significantly unidentified.

The objective of this study was to demonstrate that two commercially available products [Bovamine^®^ Dairy (*Lactobacillus animalis* LA-51 and *Propionibacterium freudenreichii* ssp. *shermani* PF-24) and Bovamine^®^ Dairy Plus (*Lactobacillus animalis* LA-51, *Propionibacterium freudenreichii* ssp. *shermani* PF-24, *Bacillus licheniformis* CH200, and *Bacillus subtilis* CH201) Chr. Hansen, Inc., Milwaukee, WI, USA], when used as daily, direct-fed microbials, are effective in reducing adverse effects of an experimentally induced *C. perfringens* infection in dairy calves.

## 2. Results

### 2.1. Performance Outcomes

**Descriptive statistics for body weight and average daily gain are shown in Table 1.** The effect of treatment on body weight gain depended significantly on study day, as depicted by a significant interaction term between treatment group and study day (*p* = 0.030; Table 2). Specifically, for all treatment groups, there was a significant (*p* < 0.001) increase in body weight on day 14 compared to day −11. On day 14, body weight gain was significantly higher for treatment 1 compared to treatment 3 (Mean difference = 11.8 kg, *p* = 0.042). Treatment was significantly (*p* = 0.050) associated with average daily gain (ADG); whereas animals in treatment 1 had a significantly higher ADG (*p* = 0.040, mean difference = 0.4 kg) than animals in treatment group 3 (Table 2).

### 2.2. Diagnostic Outcomes

**Descriptive statistics for PCR results and C.** perfringens in feces are shown in Table 3. Neither the treatment by study day interaction (*p* = 0.501), nor the main effects for treatment (*p* = 0.504) and study day (*p* = 0.469) were significantly associated with the prevalence of bacteria in feces based on the PCR assay (modeled as a dichotomous response: Yes (<40 ct), No (≥40 ct); Table 4). The effect of treatment on the concentration of bacteria among enumerable fecal samples did not depend on study day (interaction *p*-value = 0.569). When modeling main effects only, treatment was not significantly associated with the presence (*p* = 0.987) or concentration of bacteria in feces (*p* = 0.393). Study day, however, was significantly (*p* = 0.053) associated with the concentration of bacteria in feces, as concentration of bacteria in feces was significantly higher on days 0 to 7 than on days 8 to 14 (Table 4). Neither the treatment by study day interaction (*p* = 0.495), nor the main effects for treatment (*p* = 0.987) and study day (*p* = 0.569) were significantly associated with the presence of at least one CFU of bacteria in feces (modeled as a dichotomous response: Yes (≥1 CFU/g), No (0 CFU/g)).

### 2.3. Clinical Outcomes

Descriptive statistics for clinical outcome scores for diarrhea, hunger and general impression are shown in Table 5, and for skin tent/dehydration, appearance, and mortality in Table 6.

Diarrhea score (dichotomous)

The treatment by study day interaction was not significantly associated with the presence of abnormal diarrhea scores (*p* = 0.495). When evaluating main effects only, the effect of treatment (*p* = 0.005) and study day (*p* = 0.048) were both significantly associated with the presence of abnormal diarrhea scores (Table 7). Specifically, control animals had a significantly higher presence of abnormal diarrhea scores than animals in probiotic group 1 (*p* = 0.030), and probiotic group 2 (*p* = 0.005). Moreover, presence of abnormal diarrhea scores was higher on days 0 to 7 than on days 8 to 14 (*p* = 0.048).

2.Hunger scores (dichotomous)

Given the small effective sample size (hunger scores ≥ 1 = 32), a model estimating the effect of treatment over time (including main effects for treatment and study day and treatment X study day interaction), or a model including main effects only (treatment and study day) could not be fitted.

3.General impression scores (dichotomous)

The effect of treatment on the presence of abnormal general impression scores did not significantly vary by study day (interaction *p*-value = 0.086). Considering main effects only, the effect of treatment was significantly associated (*p* < 0.001) with the presence of abnormal general impression scores: control animals had a significantly higher presence of abnormal general impression scores than animals in treatment group 1 (*p* = 0.001) and treatment group 2 (*p* < 0.001; Table 7). Presence of abnormal general impression scores did not significantly vary by study day (*p* = 0.182).

4.Skin Tent scores (dichotomous)

The effect of treatment on the presence of abnormal skin tent scores did not significantly depend on study day (interaction *p*-value = 0.634). Considering main effects only, neither treatment nor study day were significantly associated (*p* > 0.05) with the presence of abnormal skin tent scores (Table 8).

5.Appearance scores (dichotomous)

The effect of treatment on the presence of abnormal appearance scores did not significantly depend on study day (interaction *p*-value = 0.253). The main effect of treatment was significantly associated (*p* < 0.001) with the presence of abnormal appearance scores, such that control calves had a significantly higher presence of abnormal appearance scores than animals in treatment group 1 (*p* = 0.003) and treatment group 2 (*p* < 0.001). Presence of abnormal appearance scores varied significantly by study day (*p* = 0.031), where presence of abnormal appearance scores was higher on days 0 to 7 than on days 8 to 14 (Table 8).

6.Mortality

**A total of 8 (26.7**%; 8/30) animals died during the entire study period. The small effective sample size prevented us from fitting multivariable models to evaluate the effect of treatment over time on mortality. Descriptive statistics for mortality by treatment group and study day are depicted in Table 6. When comparing animal-level mortality between treatment groups, 2, 0 and 6 animals died in probiotic groups 1 and 2, and controls, respectively. Based on a test of equality of proportions, mortality in control animals (60%) was significantly higher (*p* = 0.003) than mortality in animals in probiotic group 2 (0%); conversely, mortality among animals in probiotic group 1 (20%) was not significantly different than mortality among animals in probiotic group 2 (*p* = 0.136), and mortality in controls (*p* = 0.068).

## 3. Discussion

The novel findings of the present study are two-fold: (1) the ability to elicit a disease response through oral administration of a *C. perfringens* Type A to calves, and (2) the disease-mitigating effects of two commercially available probiotic products for ruminants. Calves infected with *C. perfringens* experience necro-hemorrhagic enteritis-associated diarrhea often resulting in death [7,31]; whereas, older cattle may become moribund due to enteritis and severe intraluminal hemorrhage in the jejunum [32,33] indicative of hemorrhagic jejunal syndrome (HJS) or hemorrhagic bowel syndrome (HBS) [34,35]. A study done in 2020 tested 103 fecal samples from neonatal calves, and *C. perfringens* were detected in 26 out of 103 (25.2%) neonatal calf samples [36]. From this same study, *C. perfringens* type A strains were predominant in those neonatal calves (24/26; 92.3%) [37]. Another study collected clinical samples from 227 newly born and dead diarrheic calves [36]. One hundred and forty-four of the isolates were positive for lecithinase, which indicates *C. perfringens* Type A [36]. In addition to this, 154 samples were positive by alpha toxin encoding gene-PCR assay which is responsible for the pathogenicity of *C. perfringens* Type A [36].

Experimentally induced enterotoxemia has been accomplished successfully in older calves inoculated intraduodenally with *C. perfringens* Type D [38]. Abdominal tympany, abomasitis and abomasal ulceration has been induced in calves inoculated intraruminally with toxigenic *C. perfringens* Type A [39]. Furthermore, this regimen induced anorexia, diarrhea, depression, bloat, and in some cases death. Conversely, inoculation of *C. perfringens* Type A into the abomasum or jejunum of healthy, mature, non-lactating cows failed to induce clinical signs of HJS or HBS [40], probably due to the multifactorial nature of this disease syndrome [32].

To the best of our knowledge, this is the first report of successful induction of clinical signs of disease in calves resulting from oral administration of *C. perfringens* Type A. *C. perfringens* Type A produces alpha enterotoxin. This toxin is responsible for the induction of membrane permeability alterations which damage the epithelium, allowing the enterotoxin to interact with tight junctions of the intestinal epithelial cells [41]. Damage to tight junctions disrupts the normal paracellular permeability barrier of the intestinal epithelium, which may contribute to diarrhea [41]. *Clostridium* infections can be and are being successfully treated with antibiotics, but legislation and consumer pressure toward minimizing use of antibiotics of human medicinal concern is growing; thus, biologically and economically feasible alternatives are needed [42].

The commercially available combination, in various forms, of *L. animalis* LA51 and *P. freudenreichii* PF24 probiotic bacteria has been fed to cattle in feedlots and dairy cattle since 1993 and 2003, respectively. Improvements in performance and health have been documented previously for feeding this combination probiotic in a variety of cattle types and production scenarios. When beef cattle in feedlots were fed *L. animalis* LA51 and *P. freudenreichii* PF24, ADG and feed efficiency was increased [43,44], gastrointestinal tract development was enhanced, and pathogen reductions were observed [45,46,47]. A previous study evaluated the effects of administering a live culture of *Faecalibacterium prausnitzii* to 30 newborn dairy calves on growth, health, and fecal microbiome [19]. During this study, a group of 554 Holstein heifers were assigned into treated calves (FPTRT) and non-treated calves, and the treated group presented significantly lower incidence of severe diarrhea than the control group [48]. In addition, the FPTRT group gained significantly more weight than the control group [48]. Lactating dairy cows fed this same combination of probiotic strains have responded with decreased DMI, increased milk yield, and fat- and energy-corrected milk yield [49,50,51]. Additionally, lactating cows fed *L. animalis* LA51 and *P. freudenreichii* PF24 have shown to have a favorably modified immune response system [52].

Specific to pre-weaned calves administered this combination probiotic, average ileal villus height, crypt depth, and total height (villus + crypt) were greater than non-supplemented control calves [45]. This type of small intestinal improvement was also seen in a study involving broilers. Bacillus subtilis probiotic treated broilers had reduced count of *C. perfringens* and improved the morphological status of the small intestine, as a result the feed conversation ratio also improved numerically in the group which had received the probiotic [53]. A variety of Bacilli species occur naturally in a multitude of environments; they are ubiquitous, and their inherent spore-forming properties make them ideal candidates for use as probiotics in animal feeds [54]. Bacilli spp. are easily identified in ruminant diets not supplemented with probiotics [55], diets supplemented with the Bacilli probiotic strains used in the present study [56], and fecal samples from feedlot cattle [57] further indicative of their hardiness to survive a variety of environments, including the gastrointestinal tract. Furthermore, another study found that different probiotic bacteria used in food products could inhibit *Clostridium difficile* and *Clostridium perfringens* [58]. Out of 17 commercial strains, five (2 *Lactobacillus plantarum*, 2 *Lactobacillus rhamnosus*, and 1 *Bifidobacterium animalis*) were shown to inhibit all strains of *C. difficile* and *C. perfringens* [58]. It was also discovered that two strains showed a pH-independent inhibitory effect likely due to production of either antibiotics or bacteriocins inhibiting *C. perfringens* only [58]. Based on this study, these strains have favorable growth characteristics for use as probiotics, and should be evaluated further [58]. A survey of 50 ruminal *Butyrivibrio* isolates demonstrated a high prevalence of antimicrobial production, and 26 of the 50 isolates exhibited activity against other strains of *Butyrivibrio*. These antimicrobials also showed activity against strains of *Clostridium*, *Eubacterium*, *Lachnospira*, *Lactobacillus*, *Ruminococcus*, and *Streptococcus* [59]. Other studies have shown that a lactobacilli based DFM promoted colonization of a beneficial microbiota and reduced intestinal colonization by *Clostridium perfringens* [59]. A study in 2009 evaluated a Bacillus-based direct-fed microbial and electrolytes as a therapy for scours. Fecal shedding of presumptive *Clostridium perfringens* at day 7 was reduced in scouring calves treated with electrolyte plus DFM compared to scouring calves treated with electrolytes alone. The total therapeutic treatment costs during the first two weeks were significantly reduced by supplementing the electrolyte with the DFM [60].

The aforementioned discussion of the performance and, especially, health benefits derived from feeding the specific probiotic bacteria *L. animalis* LA51 and *P. freudenreichii* PF24 to ruminants, as well as the health benefits observed previously from feeding probiotic Bacilli to poultry and swine, speak to the rationale we employed to discern the potential health benefits from feeding these probiotic combinations to dairy calves experimentally challenged with *C. perfringens* Type A. Given the limited number of calves in each feeding group, the short duration of the study and the death rates, changes in body weight of meaningful practical significance could not be determined. Notably in the present study, calves in both probiotic groups experienced significantly favorable clinical outcomes for diarrhea score and appearance score after being orally challenged with *C. perfringens*. Survival following the *C. perfringens* challenge was significantly improved when calves were fed either of the two probiotics compared with control calves. Although beyond the scope of the present study, we can only surmise that feeding dairy calves *L. animalis* LA51 and *P. freudenreichii* PF24 enhanced barrier function of the intestinal epithelium and strengthened immune response to the challenge as demonstrated previously for this probiotic combination [45,46,52]. Lastly, the fact that all the calves in probiotic group 2 (inclusion of probiotic Bacilli) survived the challenge speaks to a potentially supplemental mode of action for Bacilli versus LAB as probiotics, namely signaling interference among certain pathogens by specific strains of Bacilli. Our present findings in calves corroborate the previously published work of Van den Akker et al., 2018, who found in some studies of their meta-analysis that in randomized controlled trials with pre-term infants fed probiotics, there was a reduction in necrotizing enterocolitis, late-term sepsis, and mortality.

## 4. Materials and Methods

All activities related to this study were reviewed and approved by the Institutional Animal Care and Use Committee of Midwest Veterinary Services, Inc. prior to study initiation (IACUC number MVS18046B).

### 4.1. Animals and Study Design

Thirty (n = 30) healthy colostrum deprived dairy calves were initially selected for inclusion in the study. These calves were a day old, did not receive any vaccines or antibiotics, and all animals were born in a single day. Each calf passed an examination from a veterinarian, which deemed them to be healthy for enrollment into the study. Calves were commercially sourced from Firth, NE. The study was conducted in a randomized design. Calves were individually housed indoors on concrete floors with no nose-to-nose contact. Housing conditions were per “Guide for the Care and Use of Agricultural Cattle in Research and Teaching by the Federation of Cattle Science Societies. Individual calf was considered the experimental unit. Study personnel involved in the collection, recording or interpretation of any data were masked to the treatment assignment of cattle. The test material dispenser(s), test material administrator, and quality control personnel were unmasked to study treatments and were the only study personnel with access to the randomization and treatment assignments. Unmasked study personnel were not involved in clinical observations including recording of those observations. Calves were in overall good health with no complicating diseases reported at the time of enrollment. All calves enrolled in the study had access to veterinary care as needed. All veterinary care was at the discretion of the site veterinarian or investigator in consultation with the study monitor when possible. The study also consisted a thorough euthanasia guidelines with humane end points as per the IACUC governing bodies, veterinarians and a trained personnel. When animals meet the clinical criteria of moribund at any observation the veterinarian would intervene, and those animals would be euthanized using an AVMA approved method. Mortality within the paper would include both animals found dead, and/or euthanized; however, clostridial injections can be challenging as the disease/death can progress quickly. Due to the possible disease progression a veterinarian and/or trained staff observed the animals at minimum twice a day.

### 4.2. Testing of Probiotic Products

The study consisted of three groups of ten calves allocated randomly to three different treatments: Chr. Hansen’s probiotic 1 group (treatment 1; *Lactobacillus animalis* LA51 *and Propionibacterium freudenreichii* PF24*)*, Chr. Hansen’s probiotic 2 group (treatment 2; *L. animalis* LA51, *P. freudenreichii* PF24, *Bacillus lichenformis* CH200, *and Bacillus subtilis* CH201*)*, and control. Control calves did not receive any probiotic in the milk replacer. Probiotic 1 group received the product at an approximate dose of 3 × 10^9^ CFU per head per day in 2 g lactose throughout the study period; whereas probiotic 2 group were fed diets supplemented with probiotic at an approximate dose of 11.8 × 10^9^ CFU per head per day in 2 g lactose for the entire duration of the study. The CFU for *L. animalis* LA51 and *P. freudenreichii* PF24 in probiotic groups 1 and 2 were identical, with the total CFU difference being due to inclusion of *B. lichenformis* CH200, and *B. subtilis* CH201 in probiotic group 2. The study lasted for a period of 25 days with 4 days of acclimation (d -11 to d -7), 7 days of probiotic feeding (pre-challenge period; d -7 to d 0), oral *Clostridium* challenge (d 0), and 14 days of probiotic feeding (post-challenge; d 1 to d 14). Calves were exposed to approximately 12 h of light per day. Calves were fed twice daily a commercially available, non-medicated milk replacer (crude protein min. = 21%, crude fat min. = 20%, crude fiber max. = 0.15%, CalfCare, North Manchester, IN, USA) and received water ad libitum. Throughout the study, calves were observed twice per day and findings were recorded.

### 4.3. Experimental Challenge of Calves with Clostridium perfringens

The *C. perfringens*, Strain S107 (ATCC 13124 was available and based on preliminary challenge model development work; derived from bovine source) challenge was prepared at the CSRC, Veterinary Diagnostic Laboratory (Oakland, NE facility). The challenge material was prepared in anaerobic BHI broth. Final concentration of the challenge material was adjusted with anaerobic BHI broth to get a target dose of 1 × 10^8^ colony-forming units (CFU) per mL. The concentration of *C. perfringens* in the challenge material was performed by serial dilution (i.e., 10^−1^ to 10^−6^) in 9 mL of sterile PBS. From each dilution, 0.1 mL was spread plated on duplicate Perfringens agar plates supplemented with Kanamycin and Polymyxin B. The plates were incubated at 37 °C for 48 h in an anaerobic chamber with final counts being: Pre-challenge concentration = 1.16 × 10^8^ CFU/mL and post-challenge concentration = 9.7 × 10^7^ CFU/mL. All calves were challenged with 300 mL on day 0. This dosage was required to obtain clinical and reproducible outcome variables of interest.

### 4.4. General Health Monitoring

Routine daily observations for general health of the calves occurred during the study. Observations for clinical signs of disease associated with *Clostridial* infection included at minimum: General Health, Hunger, Skin Tent, Dehydration, Calf Appearance, and Fecal Consistency.

### 4.5. Body Weight

All calves were weighed at arrival and on conclusion of the trial. A daily scale check was performed prior to weighing cattle by placing calibrated (within the past 12 months) check weights on the scale in the following increments: 0 pounds, 50 pounds, 100 pounds, 150 pounds, and 200 pounds (1 kg = 2.2 pounds), to determine a within ±5% error. The scale weigh checks were within a ±5% error of the actual weight.

### 4.6. Blood Sample Collection

Blood samples were collected on days 7, 14 and 21 via jugular vein from all the calves. The following vacutainer method was used for blood collection, which included a 6 mL draw integrated serum separator tube (SST; COVIDien, REF # 8881302106), a vacutainer holder, and a 20-gauge × 1-inch blood collection needle (Cardinal, REF # 8881216017, Lot # 802940) for each calf. The blood collected from each calf was allowed to clot by incubation of the SST at 36 ± 2 °C for approximately one hour. The SST tubes were then centrifuged at 1400 × g for 10 min between 18–25 °C. The collected serum was aliquoted into two tubes. All serum samples were labeled with calf ID, study number, and date of collection. Serum was stored at −20 °C or colder, for subsequent analysis in the future.

### 4.7. Fecal Sample Collection

Fecal samples were collected directly from the rectum of each calf using a new glove. All samples were labeled with the calf identification, study number, and date of collection. Fecal samples were transferred to the laboratory at ambient temperature and all fecal samples were tested for *C. perfringens* using microbial plating methods and/or PCR. All fecal samples were stored at −70 °C or colder after the initial testing was performed.

### 4.8. Fecal Concentration of Clostridium perfringens

Approximately, 1 g of fecal sample from each animal was weighed and to it added 9 mL of Phosphate buffer saline (PBS). After vortexing for 30 s, a series of 10-fold dilutions were performed in PBS starting from 10^−1^ to 10^−6^ by transferring 0.1 mL of the material from tube 1 to tube 2 containing 0.9 mL of PBS. This step was repeated until 10^−6^ dilution. One hundred microliters of each dilution were plated in duplicate onto Perfringens agar plates supplemented with Kanamycin and Polymyxin B. All plates were incubated at 37 °C for approximately 48 h in an anaerobic chamber. The plates were evaluated for viable counts and the results were noted on the data capture form. The CFU/gram counts were based on the following equation:$$CFU\;per\;gram=\left(\frac{\mathrm{weight}\;\mathrm{of}\;\mathrm{fecal}\;\mathrm{sample}\;+\;\mathrm{total}\;\mathrm{volume}\;\mathrm{of}\;\mathrm{broth}\;\mathrm{added}}{Weight\;of\;fecal\;sample}\right)\times No.\;of\;colonies\;\times \;Dilution\;Factor$$

### 4.9. PCR Detection of Clostridium perfringens

DNA was extracted from each fecal sample using the Qiagen kit (QIAamp Power Fecal DNA kit), following manufacturer’s instructions. Aliquots of DNA were stored at −20 °C until PCR run. The species-specific primer set corresponding to alpha toxin gene of *C. perfringens* was used in the PCR reaction according to the published method (Selim AM et al., 2018). The PCR reaction followed by 35 cycles of 95 °C for 15 s, 56 °C for 30 s, and 72 °C for 30 s in a BioRad MyiQ thermocycler.

### 4.10. Statistical Analysis

Primary outcome variables associated with *Clostridial* infection included mortality, diarrhea, depression, dehydration, *Clostridial* fecal concentration, and body weight. Secondary outcome variables were fecal PCR results and health scores. Descriptive statistics (mean, median, standard deviation, and range) for continuous, and frequency tables for discrete outcomes, were computed by treatment group and by study day. Generalized linear mixed models (GLMM) were fitted to estimate the effect of treatment over time on production, diagnostic and clinical outcomes. Continuous outcomes such as body weight gain (kg), average daily weight gain (kg/d), and concentration of bacteria in feces among enumerable samples (concentration in log_10_ CFU/g of bacteria in feces among enumerable samples (samples with at least one CFU/g)) were modeled with a Gaussian distribution, identity link and maximum likelihood estimation. Dichotomous outcomes (yes/no; 1/0) including presence of at least one CFU of bacteria in feces, prevalence of bacteria based on PCR, and clinical scores (presence of abnormal diarrhea, hunger, general impression, skin tent and appearance scores), were modeled with a binary distribution, logit link, restricted pseudo-likelihood estimation and Kenward-Rogers degrees of freedom estimation, using PROC GLIMMIX in SAS 9.4 (SAS Institute Inc., Cary, NC, USA). The proportion of deaths (mortality) observed in each treatment group was compared using a test of equality of proportions (*pretest* in Stata 12.0; StataCorp LP., College Station, TX, USA). To estimate the effect of treatment over time on body weight gain, diagnostic and clinical outcomes, multivariable models including fixed effects for treatment group, study day and a two-way interaction term between treatment group and day were fitted. When the interaction term was not significantly associated with the outcome (*p* > 0.05), a model with main effects only (treatment group and study day) was fitted. Models included a first-order autoregressive or a heterogeneous first-order autoregressive covariance structure for animal id to account for repeated measures at the animal level (for measures equally and unequally spaced over time, respectively). A univariable model including a fixed effect for treatment was fitted to estimate the effect of treatment group on average daily gain. *p*-values < 0.05 were considered statistically significant. Means and mean percentages, standard error of the means, 95% confidence intervals and *p*-values were reported. The Tukey–Kramer adjustment for multiple comparisons was used to prevent inflation of the type I error. For interpretation of interaction terms, analyses of simple effects were computed (*slice* and *sliceby* options in *lsmeans* statement, PROC GLIMMIX). Model fit and distributional assumptions were evaluated using graphical and statistical (test) approaches.

## 5. Conclusions

*Clostridium perfringens* is a spore-forming, anaerobic bacterium which produces toxins and exoenzymes that cause disease in calves, especially necro-hemorrhagic enteritis-associated diarrhea often resulting in death. Daily feeding of *L. animalis* LA51 and *P. freudenreichii* PF24 without or with *Bacillus lichenformis* CH200 and *Bacillus subtilis* CH201, before, during and after an oral challenge of *C. perfringens* significantly reduced the incidence and severity of diarrhea while improving general impression and appearance scores of calves. Calves in the two probiotic-fed groups showed significantly higher survivability than for control calves and further substantiates the potential economic and health benefits of feeding effective probiotics.

## Figures and Tables

**Table 1 antibiotics-11-01513-t001:** Body weight and average daily gain (ADG) of dairy calves supplemented with two probiotic products to reduce *Clostridium perfringens*.

		Body Weight (kg)					ADG (kg)			
	n	Mean	Median	SD	Range	n	Mean	Median	SD	Range
Treatment										
Probiotic 1	10	87.9	85.0	15.3	64.4–116.2	10	0.9	1.0	0.4	−0.1, 1.4
Probiotic 2	10	85.5	88.2	14.2	55.6–111.0	10	0.7	0.7	0.2	0.4, 1.0
Control 3	10	81.9	81.9	10.6	63.2–100.6	10	0.5	0.5	0.4	−0.3, 1.1
Day										
−11	30	76.6	75.6	9.5	55.6–101.0	-	-	-	-	-
14	30	93.6	94.8	11.4	72.2–116.2	-	-	-	-	-

n = number of observations; SD = standard deviation.

**Table 2 antibiotics-11-01513-t002:** Body weight and average daily gain (ADG) estimated from multivariable and univariable models in dairy calves supplemented with two probiotic products to reduce *Clostridium perfringens*.

	Body Weight (in kg)					ADG (in kg)			
Variable	Mean	SEM	95% CI	*p*-value	Variable	Mean	SEM	95% CI	*p*-value
Treatment				0.350	Treatment				0.050
Probiotic 1	87.9	2.9	81.9–93.9		Probiotic 1	0.9	0.1	0.6–1.1	
Probiotic 2	85.5	2.9	79.5–91.5		Probiotic 2	0.7	0.1	0.5–1.0	
Control 3	81.9	2.9	75.9–87.8		Control 3	0.5	0.1	0.2–0.7	
Study day				<0.001			Significant contrastsSpecific to pre-weaned calves administered this combination probiotic, average ileal villus height, crypt depth, and total height (villus		
0	76.6	1.9	72.7–80.4					Mean difference	*p*-value
14	93.6	1.9	89.8–97.4				Tx 1 vs. 2	0.2	0.548
							Tx 1 vs. 3	0.4	0.040
Treatment × Study day				0.030			Tx 2 vs. 3	0.3	0.295

SEM = standard error of the mean; CI = confidence interval.

**Table 3 antibiotics-11-01513-t003:** PCR results and concentration of *C. perfringens* in feces on a continuous scale and as dichotomous outcomes (positive vs. negative) by treatment group and study day in dairy calves supplemented with two probiotic products to reduce *Clostridium perfringens*.

		PCR Fecal (in Ct Values ^†^) (n = 193)					Concentration of Bacteria in Feces (in CFU/g) (n = 193)			
	n	Mean	Median	SD	Range		Mean	Median	SD	Range
Treatment										
Probiotic 1	66	37.7	40.0	5.2	22.5–40.0		257,924.2	950.0	1,257,737.2	0–100,000,000
Probiotic 2	70	36.4	40.0	6.5	19.5–40.0		1,683,386.4	1650.0	7,961,623.4	0–60,000,000
Control 3	57	35.9	40.0	7.0	20.1–40.0		651,790.4	3850.0	2,484,994.3	0–14,500,000
Day										
0	30	40.0	40.0	0.0	40.0–40.0		0.0	0.0	0.0	0.0–0.0
1	30	34.3	40.0	6.9	20.1–40.0		1,142,935.0	16,000.0	3,217,369.2	50–14,500,000.0
2	30	37.4	40.0	5.9	21.7–40.0		894,460.0	31,500.0	2,330,707.9	0–12,000,000.0
3	30	36.1	40.0	6.6	20.5–40.0		2,290,015.0	19,750.0	10,914,622.9	0–60,000,000.0
4	26	34.4	40.0	7.9	19.5–40.0		1,394,369.2	4050.0	5,859,102.3	0–30,000,000.0
7	25	36.1	40.0	6.7	20.8–40.0		235,226.0	16,500.0	508,041.9	0–2,000,000.0
14	22	38.5	40.0	5.0	21.6–40.0		2525.0	125.0	7091.6	0–29,000.0
	Prevalence based on PCR (dichotomous)						Presence of at least one CFU of bacteria in feces (dichotomous)			
	n	Pos (<40)	Neg (≥40)			n	Pos (≥1 CFU/g)	Neg (0 CFU/g)		
Treatment										
Probiotic 1	66	12	54			66	42	24		
Probiotic 2	70	18	52			70	44	26		
Control 3	57	16	41			57	35	22		
Total n (%)	193	46 (23.8)	147 (76.2)			193	121 (62.7)	72 (37.3)		
	n	Pos (<40)	Neg (≥40)				Pos (≥1 CFU/g)	Neg (0 CFU/g)		
Day										
0	30	0	30				0	30		
1	30	14	16				30	0		
2	30	5	25				24	6		
3	30	9	21				23	7		
4	26	9	17				14	12		
7	25	7	18				19	6		
14	22	2	20				11	11		

^†^ Negative samples were assigned a value of 40 ct; n = number of observations; SD = standard deviation.

**Table 4 antibiotics-11-01513-t004:** Prevalence of *C. perfringens* in feces based on PCR (dichotomous outcome) and concentration of bacteria in feces (modeled as dichotomous and continuous outcome) estimated from multivariable models ^a^.

Variable	Prevalence of *C. perfringens* in Feces Based on the PCR Assay				Variable	Presence of at Least One CFU of *C. perfringens* in Feces (n = 193 samples)			
	Mean %	SEM	Mean % 95% CI	*p*-Value		Mean %	SEM	Mean % 95% CI	*p*-Value
Treatment				0.504	Treatment				0.987
Probiotic 1	16.5	5.3	8.4–29.9		Probiotic 1	63.1	7.1	48.2–75.8	
Probiotic 2	23.5	6.0	13.7–37.3		Probiotic 2	62.0	6.9	47.7–74.5	
Control 3	25.6	7.0	14.2–41.6		Control 3	61.5	7.8	45.2–75.5	
Study day				0.469	Study day				0.569
Day 0 to 7	24.5	4.0	17.4–33.3		Day 0 to 7	59.6	4.6	50.2–68.4	
Day 8 to 14	18.9	6.2	9.5–34.3		Day 8 to 14	64.7	7.6	48.7–77.9	

^a^ Multivariable model estimating the effect of treatment and study day on diagnostic outcomes (each outcome modeled separately) included fixed effects for treatment group and study day and a covariance structure.

**Table 5 antibiotics-11-01513-t005:** Diarrhea, hunger, and general impression scores by treatment group and study day in dairy calves supplemented with two probiotic products to reduce *Clostridium perfringens*.

		Diarrhea Scores				Hunger Scores				General Impression Scores				
	n	0	1	2	3	0	1	2	3	0	1	2	3	4
Treatment														
Probiotic 1	132	87	40	4	1	123	6	2	1	101	24	5	0	2
Probiotic 2	150	108	38	4	0	149	1	0	0	125	22	3	0	0
Control 3	98	31	25	26	16	76	11	8	3	39	24	21	11	3
Total n (%)	380	226 (59.5)	103 (27.1)	34 (8.9)	17 (4.5)	348 (91.6)	18 (4.7)	10 (2.6)	4 (1.1)	265 (69.8)	70 (18.4)	29 (7.6)	11 (2.9)	5 (1.3)

n = number of observations.

**Table 6 antibiotics-11-01513-t006:** Dehydration and appearance scores and animal level mortality by treatment group and study day in dairy calves supplemented with two probiotic products to reduce *Clostridium perfringens*.

		Skin Tent/Dehydration Scores					Appearance Scores					Mortality		Mortality (Animal Level)
	n	0	1	2	3	4	0	1	2	3	4	No	Yes	Proportion (%)
Treatment														
Probiotic 1	132	86	42	3	0	1	102	25	3	0	2	130	2	2/10 (20.0)
Probiotic 2	150	109	41	0	0	0	124	26	0	0	0	150	0	0/10 (0.0)
Control 3	98	43	19	25	9	2	35	23	28	9	3	92	6	6/10 (60.0)
Total n (%)	380	238 (62.6)	102 (26.8)	28 (7.4)	9 (2.4)	3 (0.8)	261 (68.6)	74 (19.5)	31 (8.2)	9 (2.4)	5 (1.3)	372 (97.9)	8 (2.1)	8/30 (26.7)

n = number of observations.

**Table 7 antibiotics-11-01513-t007:** Diarrhea and general impression scores modeled as dichotomous outcomes estimated from multivariable models ^a^ among dairy calves supplemented with two probiotic products to reduce *Clostridium perfringens*.

	Diarrhea Score (≥1 vs. 0)				General Impression (≥1 vs. 0)			
Variable	Mean %	SEM	Mean 95% CI	*p*-Value	Mean %	SEM	Mean 95% CI	*p*-Value
Treatment				0.005				<0.001
Probiotic 1	29.3	6.4	18.3–43.4		21.4	5.1	13.0–33.2	
Probiotic 2	23.5	5.6	14.2–36.4		15.0	4.2	8.4–25.2	
Control 3	57.0	8.1	40.7–71.9		55.1	7.2	40.7–68.7	
			*Significant contrasts*				*Significant contrasts*	
			3 vs. 1	0.030			3 vs. 1	<0.001
			3 vs. 2	0.005			3 vs. 2	<0.001
Study day				0.048				0.182
Day 0 to 7	43.01	5.2	33.2–53.6		32.7	4.7	24.2–42.5	
Day 8 to 14	28.8	5.6	19.0–40.9		23.8	5.0	15.3–35.0	

^a^ Multivariable model estimating the effect of treatment and study day on clinical outcomes (each outcome modeled separately), included fixed effects for treatment group and study day, and a covariance structure.

**Table 8 antibiotics-11-01513-t008:** Skin tent and appearance scores modeled as dichotomous outcomes estimated from multivariable models ^a^ among dairy calves supplemented with two probiotic products to reduce *Clostridium perfringens*.

	Skin Tent (≥1 vs. 0)				Appearance (≥1 vs. 0)			
Variable	Mean %	SEM	Mean 95% CI	*p*-Value	Mean %	SEM	Mean 95% CI	*p*-Value
Treatment				0.145				<0.001
Probiotic 1	29.5	7.6	16.8–46.5		19.4	5.2	11.1–31.8	
Probiotic 2	22.3	6.5	11.9–37.9		14.5	4.3	7.8–25.4	
Control 3	45.0	9.4	27.7–63.6		54.2	7.7	38.9–68.7	
							*Significant contrasts*	
							3 vs. 1	0.003
							3 vs. 2	<0.001
Study day				0.526				0.031
Day 0 to 7	33.9	5.5	23.8–45.6		34.7	4.9	25.6–45.0	
Day 8 to 14	29.4	6.1	18.8–42.8		20.0	4.8	12.1–31.2	

^a^ Multivariable model estimating the effect of treatment and study day on clinical outcomes (each outcome modeled separately), included fixed effects for treatment group and study day and a covariance structure.

## Data Availability

The data presented in this study are available upon reasonable request from the corresponding author.

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
