# Peer review of "Efficacy of Two Probiotic Products Fed Daily to Reduce Clostridium perfringens-Based Adverse Health and Performance Effects in Dairy Calves"

_antibiotics, 2022, doi:10.3390/antibiotics11111513_

Round 1
Reviewer 1 Report
Antibiotics 10.3390 “Efficacy of two probiotic products fed daily to reduce Clostridium perfringens-based adverse health and performance effects in dairy calves”
I have two main concerns regarding this manuscript:
1) it appears that death was used as an experimental endpoint with no indication of any attempt to intervene and euthanize the morbid calves.
2) the n values in the tables to not add up and the data is presented across the three treatments making it useless and unable to evaluate the experimental results.
Guidelines for Endpoints in Animal Study Proposals (nih.gov)
Office of Laboratory Animal Welfare; Institutional Animal Care and Use Committee Guidebook (nih.gov)
OLAW Institutional Animal Care and Use Committee Guidebook 2002 2nd edition
104 C. Review of Proposals
Moribund Condition as a Humane Endpoint:
Moribund has been defined as “in the state of dying,” or “at the point of death.” A moribund condition may be an appropriate humane experimental endpoint for some studies where there is the induction of severe disease states and high rates of mortality. Pre-emptive euthanasia of moribund animals can prevent further pain and distress. Objective data-based criteria that are predictive of impending death can be used to implement timely euthanasia to avoid spontaneous deaths. FDA regulatory testing guidelines allow for humane killing of animals that are moribund. However, it is important to recognize that euthanasia of a moribund animal does not eliminate pain and distress that may be experienced during progression to a moribund condition. It should also be noted that while death is not a required endpoint for routine toxicity testing, animals are often found dead during studies. Establishing procedures to detect and humanely euthanize moribund or pre-moribund animals can reduce the number of animals that die spontaneously. In addition to reducing animal pain and distress, euthanasia of moribund animals allows for the collection of tissues and other biologic specimens that may otherwise be lost or rendered unusable when an animal is found dead. C.2. Protocol Review Criteria 105 Various clinical signs are indicative of a moribund condition in laboratory animals. These typically include one or more of the following: • impaired ambulation which prevents animals from reaching food or water, • excessive weight loss and emaciation, • lack of physical or mental alertness, • difficult labored breathing, and • inability to remain upright. Animals should be observed frequently enough to detect signs of impending death so they can be euthanized in a timely manner. When increased morbidity or mortality is expected, a minimum of twice daily observation is recommended. Animals not likely to survive until the next scheduled observation should normally be euthanized. In situations where animals are often found dead, closer and more frequent observation for moribund animals should be considered to reduce spontaneous deaths. Euthanasia of animals that are moribund or experiencing severe pain and distress should always be done in a manner that produces the least possible amount of additional pain and distress. Other Humane Endpoints in Research Animals used to study tumor biology, to develop new cancer therapies, and to evaluate the carcinogenic potential of substances may experience pain and distress. Frequent and appropriate monitoring of animals during tumor development is necessary to allow for appropriate intervention before significant deterioration or death. Effective monitoring systems and endpoints should include limits on tumor size and severity of tumor-associated disease. Altered physiologic, biochemical, and other biomarkers may be potentially more objective and reproducible endpoints than clinical signs for such studies. Genetically engineered animal models are sometimes accompanied by unintended and unpredicted alterations that adversely affect animal wellbeing. Investigators need to establish a plan for addressing unanticipated adverse outcomes for genetically altered animals. There should be a plan for systematic characterization of phenotypes to facilitate assessment of their possible utility and timely decisions on disposition or retention. IACUCs should provide oversight of such studies to ensure that animal welfare problems are handled in an effective and prompt manner. 106 C. Review of Proposals Animals with induced infections may experience significant pain and/or distress during progression of the disease. Early physiologic and biochemical changes during infection have been found to be useful humane endpoints rather than death or moribund condition. Specific decreases in body temperature have been found to be effective early predictors of eventual death for some infections in rodents. Vaccine potency testing typically involves challenging immunized animals with infectious agents. While such testing has commonly used lethality as the endpoint indicative of insufficient protection, some regulatory authorities now allow euthanasia of moribund animals.
In L26-32 The authors stated that daily observations were made
“Daily feeding of L. animals LA51 and P. freudenreichii PF24 without or with Bacillus lichenformis CH200 and Bacillus subtilis CH201, before, during and after an oral challenge of C. perfringens significantly reduced the incidence and severity of diarrhea while improving general impression and appearance scores of calves. Most notably, survival of calves in the two probiotic-fed groups was significantly higher than for control calves and further substantiates the potential economic and health benefits of feeding effective probiotics.”
Also, in L408 “Throughout the study, calves were observed twice per day and findings were recorded.”
The authors should have intervened with euthanasia prior to the deaths on days 5, 7, and 9.
Specific comments:
L60 Provide a reference for this statement related to treating calves with Clostridia infections. “In addition, alternatives for disease prevention in animals are being considered due to the European ban of antibiotics in livestock.”
L385 Delete second repetitive statement “Calves had never been treated with an antibiotic or vaccine prior to initiating the study” since it repeats information provided in L372 “These calves were a day old, did not receive any vaccines or antibiotics”.
L423 Provide a reference for this statement “This dosage was required to obtain clinical and reproducible outcome variables of interest.”
L432 Provide the rational for weighing calves at arrival (D-11) and on conclusion of the trial (D14) and not every 7 or less days.
L439 Explain how calves were blood sampled on day 21 when the experiment only lasted to day 14.
L448 Provide the frequency that fecal samples were collected.
Tables:
The number in Tables 5 and 6 do not add-up to the n values for observation on each day.
The data in all tables should be presented by treatment group.
Table 1 that data is not presented in a usable format. Both body weight and ADG are presented for the entire 25 days of the study from day -11 to day 14, but for Treatments 1 and 3 the number calves was not n=10 due to mortality.
Since mortality occurred on days 4, 5, 7, and 9, it is not possible to have n=30 for day 14.
Table 3 the PCR “Day” results should be presented by treatment group not the average for the 3 treatment groups.
In Table 5 Explain why the “n” values differ on days 2, 3, 4, 5, 6, 7, 8, and 9.
Day n total
2 28 30
3 24 30
4 18 26 27 26
5 20 26
6 22 26
7 22 25
8 21 24
9 23 24
10 22 22
11 22 22
12 22 22
13 22 22
14 22 22
In Table 6 again the n values do not add up for Skin tent and Appearance.
Author Response
I have two main concerns regarding this manuscript:
- it appears that death was used as an experimental endpoint with no indication of any attempt to intervene and euthanize the morbid calves.
Agreed and complied. We have revised the ‘Animals and Study design’ section accordingly.
- the n values in the tables to not add up and the data is presented across the three treatments making it useless and unable to evaluate the experimental results.
Agreed and complied. We have revised the tables accordingly.
Guidelines for Endpoints in Animal Study Proposals (nih.gov)
Office of Laboratory Animal Welfare; Institutional Animal Care and Use Committee Guidebook (nih.gov)
OLAW Institutional Animal Care and Use Committee Guidebook 2002 2nd edition
104 C. Review of Proposals
Moribund Condition as a Humane Endpoint:
Moribund has been defined as “in the state of dying,” or “at the point of death.” A moribund condition may be an appropriate humane experimental endpoint for some studies where there is the induction of severe disease states and high rates of mortality. Pre-emptive euthanasia of moribund animals can prevent further pain and distress. Objective data-based criteria that are predictive of impending death can be used to implement timely euthanasia to avoid spontaneous deaths. FDA regulatory testing guidelines allow for humane killing of animals that are moribund. However, it is important to recognize that euthanasia of a moribund animal does not eliminate pain and distress that may be experienced during progression to a moribund condition. It should also be noted that while death is not a required endpoint for routine toxicity testing, animals are often found dead during studies. Establishing procedures to detect and humanely euthanize moribund or pre-moribund animals can reduce the number of animals that die spontaneously. In addition to reducing animal pain and distress, euthanasia of moribund animals allows for the collection of tissues and other biologic specimens that may otherwise be lost or rendered unusable when an animal is found dead. C.2. Protocol Review Criteria 105 Various clinical signs are indicative of a moribund condition in laboratory animals. These typically include one or more of the following: • impaired ambulation which prevents animals from reaching food or water, • excessive weight loss and emaciation, • lack of physical or mental alertness, • difficult labored breathing, and • inability to remain upright. Animals should be observed frequently enough to detect signs of impending death so they can be euthanized in a timely manner. When increased morbidity or mortality is expected, a minimum of twice daily observation is recommended. Animals not likely to survive until the next scheduled observation should normally be euthanized. In situations where animals are often found dead, closer and more frequent observation for moribund animals should be considered to reduce spontaneous deaths. Euthanasia of animals that are moribund or experiencing severe pain and distress should always be done in a manner that produces the least possible amount of additional pain and distress. Other Humane Endpoints in Research Animals used to study tumor biology, to develop new cancer therapies, and to evaluate the carcinogenic potential of substances may experience pain and distress. Frequent and appropriate monitoring of animals during tumor development is necessary to allow for appropriate intervention before significant deterioration or death. Effective monitoring systems and endpoints should include limits on tumor size and severity of tumor-associated disease. Altered physiologic, biochemical, and other biomarkers may be potentially more objective and reproducible endpoints than clinical signs for such studies. Genetically engineered animal models are sometimes accompanied by unintended and unpredicted alterations that adversely affect animal wellbeing. Investigators need to establish a plan for addressing unanticipated adverse outcomes for genetically altered animals. There should be a plan for systematic characterization of phenotypes to facilitate assessment of their possible utility and timely decisions on disposition or retention. IACUCs should provide oversight of such studies to ensure that animal welfare problems are handled in an effective and prompt manner. 106 C. Review of Proposals Animals with induced infections may experience significant pain and/or distress during progression of the disease. Early physiologic and biochemical changes during infection have been found to be useful humane endpoints rather than death or moribund condition. Specific decreases in body temperature have been found to be effective early predictors of eventual death for some infections in rodents. Vaccine potency testing typically involves challenging immunized animals with infectious agents. While such testing has commonly used lethality as the endpoint indicative of insufficient protection, some regulatory authorities now allow euthanasia of moribund animals.
Thank you!
In L26-32 The authors stated that daily observations were made
“Daily feeding of L. animals LA51 and P. freudenreichii PF24 without or with Bacillus lichenformis CH200 and Bacillus subtilis CH201, before, during and after an oral challenge of C. perfringens significantly reduced the incidence and severity of diarrhea while improving general impression and appearance scores of calves. Most notably, survival of calves in the two probiotic-fed groups was significantly higher than for control calves and further substantiates the potential economic and health benefits of feeding effective probiotics.”
Also, in L408 “Throughout the study, calves were observed twice per day and findings were recorded.”
The authors should have intervened with euthanasia prior to the deaths on days 5, 7, and 9.
Agreed and complied. We have revised the ‘Animals and Study design’ section accordingly.
Specific comments:
L60 Provide a reference for this statement related to treating calves with Clostridia infections. “In addition, alternatives for disease prevention in animals are being considered due to the European ban of antibiotics in livestock.”
Agreed and complied.
L385 Delete second repetitive statement “Calves had never been treated with an antibiotic or vaccine prior to initiating the study” since it repeats information provided in L372 “These calves were a day old, did not receive any vaccines or antibiotics”.
Agreed and Complied.
L423 Provide a reference for this statement “This dosage was required to obtain clinical and reproducible outcome variables of interest.”
We did standardize the procedure by conducting a pilot study first.
L432 Provide the rational for weighing calves at arrival (D-11) and on conclusion of the trial (D14) and not every 7 or less days.
This study was focused on the overall body weight affects that an end producer may see without biasing the performance affect with interim cattle handling/stress of collecting body weights.
L439 Explain how calves were blood sampled on day 21 when the experiment only lasted to day 14.
This section needs updated as this was a typo and should have been study day 14. Raghu, this blood collection section should be removed from the material and methods as we are not illustrating any blood results for this paper.
L448 Provide the frequency that fecal samples were collected.
Fecal samples were collected on Study Day -11, 0 (prior to challenge), 1, 2, 3, 4, 7, and 14. If an animal is found dead or was euthanized prior to the end of the study a fecal sample will be collected.
Tables:
The number in Tables 5 and 6 do not add-up to the n values for observation on each day.
Agreed and Complied.
The data in all tables should be presented by treatment group.
Agreed and Complied.
Table 1 that data is not presented in a usable format. Both body weight and ADG are presented for the entire 25 days of the study from day -11 to day 14, but for Treatments 1 and 3 the number calves was not n=10 due to mortality.
Since mortality occurred on days 4, 5, 7, and 9, it is not possible to have n=30 for day 14.
Agreed and Complied.
Table 3 the PCR “Day” results should be presented by treatment group not the average for the 3 treatment groups.
We have revised accordingly and presented the data by treatment group (overall). And also, we have provided a table by day.
In Table 5 Explain why the “n” values differ on days 2, 3, 4, 5, 6, 7, 8, and 9.
Day n total
2 28 30
3 24 30
4 18 26 27 26
5 20 26
6 22 26
7 22 25
8 21 24
9 23 24
10 22 22
11 22 22
12 22 22
13 22 22
14 22 22
We have revised accordingly and presented the data by treatment group (overall).
In Table 6 again the n values do not add up for Skin tent and Appearance.
Agreed and Complied.
Reviewer 2 Report
The manuscript by Cull et al. aimed to demonstrate the efficacy of two probiotic products in reducing the adverse effects of experimental Clostridium perfringens infection in dairy calves. The authors concluded that daily feeding of probiotics, before, during and after an oral challenge of C. perfringens, significantly reduces the incidence and severity of diarrhea, in addition to improving the survival of animals in the groups fed with probiotics. The question investigated in the manuscript is important and relevant. I did a survey and didn't find other studies that evaluate these products in the attenuation of infection by C. perfringens (producer of alpha enterotoxin). The text is clear but must undergo a style review, but I believe it can be published after that. Below are some minor comments:
- The introduction and discussion sections were presented as a single, long paragraph each. Please review them;
- When citing the two probiotic products for the first time (end of the introduction), I suggest that the authors present the composition of each one and the full name of the microorganisms (since the species have not yet been mentioned);
- Many bacteria names are not in italics, check the entire manuscript;
- Some references have authors in bold or are interrupted. Please, the author should check the MDPI guidelines;
- Lines 247 – 253: abbreviations must be placed after the conclusions (confirm this in the MDPI guidelines).
Author Response
The manuscript by Cull et al. aimed to demonstrate the efficacy of two probiotic products in reducing the adverse effects of experimental Clostridium perfringens infection in dairy calves. The authors concluded that daily feeding of probiotics, before, during and after an oral challenge of C. perfringens, significantly reduces the incidence and severity of diarrhea, in addition to improving the survival of animals in the groups fed with probiotics. The question investigated in the manuscript is important and relevant. I did a survey and didn't find other studies that evaluate these products in the attenuation of infection by C. perfringens (producer of alpha enterotoxin). The text is clear but must undergo a style review, but I believe it can be published after that. Below are some minor comments:
Thank you!
- The introduction and discussion sections were presented as a single, long paragraph each. Please review them;
Agreed and complied.
- When citing the two probiotic products for the first time (end of the introduction), I suggest that the authors present the composition of each one and the full name of the microorganisms (since the species have not yet been mentioned);
Agreed and revised accordingly.
Treatment 1
“8 calves survived the challenge…”
Bovamine Dairy, daily feeding rate of 3.0x10E09 cfu
Lactobacillus animalis (LA-51)
Propionibacterium freudenreichii ssp. shermanii (PF-24)
Treatment 2
“10 calves survived the challenge…”
Bovamine Dairy Plus, daily feeding rate of 11.8x10E09 cfu
Lactobacillus animalis (LA-51)
Propionibacterium freudenreichii ssp. shermanii (PF-24)
Bacillus licheniformis (CH200)
Bacillus subtilis (CH201)
- Many bacteria names are not in italics, check the entire manuscript;
Agreed and complied.
- Some references have authors in bold or are interrupted. Please, the author should check the MDPI guidelines;
We have revised the references. Thank you!
- Lines 247 – 253: abbreviations must be placed after the conclusions (confirm this in the MDPI guidelines).
Agreed and complied.
Round 2
Reviewer 1 Report
Review Antibiotics 10.3390
"Efficacy of two probiotic products fed daily to reduce Clostridium perfringens-based adverse health and performance effects 3 in dairy calves"
The tables are much improved but the combining of calves found dead and/or potentially euthanasia is not acceptable. This research appears to have used death as an endpoint and stated as such in L 29-30 “Most notably, survival of calves in the two probiotic-fed groups was significantly higher than for control calves and further substantiates the potential economic and health benefits of feeding effective probiotics.” and in L548-550 “Most notably, survival of calves in the two probiotic-fed groups was significantly higher than for control calves and further substantiates the potential economic and health benefits of feeding effective probiotics.”
This manuscript in L282 states “The novel findings of the present study are two-fold: 1) the ability to elicit a disease response through oral administration of a C. perfringens Type A to calves,”
And in L439 “The C. perfringens, Strain S107 (ATCC 13124 was available and based on preliminary challenge model development work; derived from bovine source) challenge was prepared at the CSRC, Veterinary Diagnostic Laboratory (Oakland, NE facility).”
“final counts being: Pre-challenge concentration = 1.16 x 10^8 CFU/mL and post-challenge concentration = 9.7 x 10^7 CFU/mL. All calves were challenged with 300 mL on day 0. This dosage was required to obtain clinical and reproducible outcome variables of interest.”
Specific comments:
For Tables 1, 2, 3, and 4 the time period day -11 to +14 need to be added, for a total of 25 days.
Tables 3, 5, and 6 L153, delete “100%” for the Total n.
L97 Insert “kg”, …Mean differences = 11.8 kg,…
Table 6 and in L291 Delete values for mortality to the right of the decimal point when the sample number is less than 100.
L332 Insert “villus”, …average ileal villus height,…
L 332-338 For both reference #1 and #11 an increase in villus height or crypt depth accompanied with no significant difference in final body weight or carcass yield compared to controls indicates decreased efficiency with a speculated increased surface area. Also, villi are 3 dimensional and villi height alone does not account for villi total surface area.
Reference #11. During the growing period, there were no statistical differences in BW, ADG, feed intake, or G:F. Average daily gain showed a trend for being greater (quadratic; P < 0.07) for Holstein steers receiving 1 × 10^5 L. acidophilus versus control steers and steers receiving 1 × 10^6 L. acidophilus during the finishing period. No differences (P > 0.10) were noted in daily intake or G:F between treatments during the finishing period. Overall, steers fed 1 × 10^5 L. acidophilus had no difference in final BW and ADG versus steers fed the control diet.
Reference #1. ROV supplementation had a positive effect on physical alterations in the structure of the small intestine but it was without effect on BWG and FCR.
No improvement in carcass characteristics was reported in this experiment which appeared in parallel to the BWG result. Similarly, Mushtaq et al. 16 reported that breast meat yield was not affected
by the addition of ROV Excel enzyme to broilers diet. Yeon et al. 19 reported enzyme supplementation did not improve the relative weights of the leg or breast muscle.
L412 delete “humanly” since euthanasia by definition is humane, euthanasia that is not humane is either killing or slaughter.
Author Response
The tables are much improved but the combining of calves found dead and/or potentially euthanasia is not acceptable. This research appears to have used death as an endpoint and stated as such in L 29-30 “Most notably, survival of calves in the two probiotic-fed groups was significantly higher than for control calves and further substantiates the potential economic and health benefits of feeding effective probiotics.” and in L548-550 “Most notably, survival of calves in the two probiotic-fed groups was significantly higher than for control calves and further substantiates the potential economic and health benefits of feeding effective probiotics.”
Agreed and Complied. We have slightly revised the sentence.
This manuscript in L282 states “The novel findings of the present study are two-fold: 1) the ability to elicit a disease response through oral administration of a C. perfringens Type A to calves,”
And in L439 “The C. perfringens, Strain S107 (ATCC 13124 was available and based on preliminary challenge model development work; derived from bovine source) challenge was prepared at the CSRC, Veterinary Diagnostic Laboratory (Oakland, NE facility).”
“final counts being: Pre-challenge concentration = 1.16 x 10^8 CFU/mL and post-challenge concentration = 9.7 x 10^7 CFU/mL. All calves were challenged with 300 mL on day 0. This dosage was required to obtain clinical and reproducible outcome variables of interest.”
Specific comments:
Agreed and this is the strain we used to test our hypothesis. Thank you!
For Tables 1, 2, 3, and 4 the time period day -11 to +14 need to be added, for a total of 25 days.
Agreed and Complied.
Tables 3, 5, and 6 L153, delete “100%” for the Total n.
Agreed and Complied.
L97 Insert “kg”, …Mean differences = 11.8 kg,…
Agreed and Complied.
Table 6 and in L291 Delete values for mortality to the right of the decimal point when the sample number is less than 100.
Just to be consistent, we would like to retain this mortality information. Thank you!
L332 Insert “villus”, …average ileal villus height,…
Agreed and Complied.
L 332-338 For both reference #1 and #11 an increase in villus height or crypt depth accompanied with no significant difference in final body weight or carcass yield compared to controls indicates decreased efficiency with a speculated increased surface area. Also, villi are 3 dimensional and villi height alone does not account for villi total surface area.
Agreed. But, this is what we could be best capture and extrapolate our results to previously published work.
Reference #11. During the growing period, there were no statistical differences in BW, ADG, feed intake, or G:F. Average daily gain showed a trend for being greater (quadratic; P < 0.07) for Holstein steers receiving 1 × 10^5 L. acidophilus versus control steers and steers receiving 1 × 10^6 L. acidophilus during the finishing period. No differences (P > 0.10) were noted in daily intake or G:F between treatments during the finishing period. Overall, steers fed 1 × 10^5 L. acidophilus had no difference in final BW and ADG versus steers fed the control diet.
Agreed and Complied.
Reference #1. ROV supplementation had a positive effect on physical alterations in the structure of the small intestine but it was without effect on BWG and FCR.
No improvement in carcass characteristics was reported in this experiment which appeared in parallel to the BWG result. Similarly, Mushtaq et al. 16 reported that breast meat yield was not affected
by the addition of ROV Excel enzyme to broilers diet. Yeon et al. 19 reported enzyme supplementation did not improve the relative weights of the leg or breast muscle.
L412 delete “humanly” since euthanasia by definition is humane, euthanasia that is not humane is either killing or slaughter.
Agreed and Complied.